

# Harmonic chain far from equilibrium: Single-file diffusion, long-range order, and hyperuniformity

**Harukuni Ikeda**

Department of Physics, Gakushuin University, 1-5-1 Mejiro,
Toshima-ku, Tokyo 171-8588, Japan

[harukuni.ikeda@gakushuin.ac.jp](mailto:harukuni.ikeda@gakushuin.ac.jp)

## Abstract

In one dimension, particles can not bypass each other. As a consequence, the mean-squared displacement (MSD) in equilibrium shows sub-diffusion $\text{MSD}(t) \sim t^{1/2}$, instead of normal diffusion $\text{MSD}(t) \sim t$. This phenomenon is the so-called single-file diffusion. Here, we investigate how the above equilibrium behaviors are modified far from equilibrium. In particular, we want to uncover what kind of non-equilibrium driving force can suppress diffusion and achieve the long-range crystalline order in one dimension, which is prohibited by the Mermin-Wagner theorem in equilibrium. For that purpose, we investigate the harmonic chain driven by the following four types of driving forces that do not satisfy the detailed balance: (i) temporally correlated noise with the noise spectrum $D(\omega) \sim \omega^{-2\theta}$, (ii) conserving noise, (iii) periodic driving force, and (iv) periodic deformations of particles. For the driving force (i) with $\theta > -1/4$, we observe $\text{MSD}(t) \sim t^{1/2+2\theta}$ for large $t$. On the other hand, for the driving forces (i) with $\theta < -1/4$ and (ii)-(iv), MSD remains finite. As a consequence, the harmonic chain exhibits the crystalline order even in one dimension. Furthermore, the density fluctuations of the model are highly suppressed in a large scale in the crystal phase. This phenomenon is known as hyperuniformity. We discuss that hyperuniformity of the noise fluctuations themselves is the relevant mechanism to stabilize the long-range crystalline order in one dimension and yield hyperuniformity of the density fluctuations.

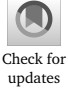
doi:[10.21468/SciPostPhys.17.4.103](https://doi.org/10.21468/SciPostPhys.17.4.103)

# 1 Introduction

In one-dimensional many-particle systems, the particles cannot bypass one another if the interactions are strong enough. As a consequence, the mean-squared displacement (MSD) in equilibrium shows sub-diffusion $\text{MSD}(t) \sim t^{1/2}$ [1–6], instead of normal diffusion $\text{MSD}(t) \sim t$ [7]. This phenomenon is known as single-file diffusion. The simplest model to observe single-file diffusion is the one-dimensional harmonic chain, where point-like particles on a line are connected by harmonic springs [1]. The harmonic chain is often recognized as a toy model of a one-dimensional crystal [8, 9]. However, as proved by Mermin and Wagner, the long-range order cannot exist in one and two dimensions in equilibrium if interactions are short-ranged [9–11]. This implies that the particles will diffuse away from their lattice positions after a sufficiently long time. As a consequence, MSD of the harmonic chain grows as $\text{MSD} \sim t^{1/2}$, as in the case of standard single-file diffusion [1]. The aim of this manuscript is to discuss how the above equilibrium behaviors are changed if the model is driven by athermal fluctuations violating the detailed balance. In particular, we show that for specific types of athermal fluctuations, the diffusion is strongly suppressed, and as a consequence, the harmonic chain can have the long-range crystalline order even in one dimension.

Our model also provides an ideal playground to investigate hyperuniformity far from equilibrium. Hyperuniformity is a phenomenon that the large-scale fluctuations of physical quantities are anomalously suppressed. In particular, hyperuniformity of the density fluctuations is characterized by the vanishing of the static structure factor $S(q)$ in the limit of the small wave number $q$: $\lim_{q \to 0} S(q) = 0$ [12]. Hyperuniformity has been observed in perfect crystals at zero temperature [13], quasicrystals [14, 15], ground states of quantum systems [16–18], periodically driven emulsions [19], chiral active matter [20–23], and so on [24, 25]. Interestingly, a recent numerical study reported that hyperuniformity of out-of-equilibrium systems can also suppress the critical fluctuations and stabilize the crystaline order even in two dimension [26], which is prohibited by the Mermin-Wagner theorem in equilibrium [10, 11]. So far, most of the theoretical studies of hyperuniformity far from equilibrium have been conducted on low densities much below the crystal phase [20, 23, 27]. We believe that the harmonic chain plays the role of the minimal model for investigating how hyperuniformity appears and stabilizes the crystalline order in low-dimensional systems far from equilibrium.

The manuscript is organized as follows. In Sec. 2, we introduce the model and define a few important physical quantities. Then, we perform case studies for the following four types of driving forces that do not satisfy the detailed balance.

Firstly, in Sec. 3, we consider the temporally correlated noise with the power-law noise spectrum $D_q(\omega) \sim \omega^{-2\theta}$. Although the model may appear somewhat artificial, it allows us to systematically investigate how the temporal correlations enhance or suppress the diffusion and yield long-range crystalline order and hyperuniformity by continuously changing the value of $\theta$. We show that for $\theta > -1/4$, the mean-squared displacement behaves as MSD $\sim t^{1/2+2\theta}$ for large $t$. For $\theta < -1/4$, on the contrary, the diffusion is completely suppressed, and MSD converges to a finite value in the long time limit. As a consequence, the model exhibits the long-range crystalline order even in one dimension, which is prohibited in equilibrium [10, 11]. Furthermore, we show that the static structure factor $S(q)$ for a small wave number $q$ behaves as $S(q) \sim q^{-4\theta}$ in the crystal phase. In the crystal phase, $S(q) \to 0$ in the limit $q \to 0$, meaning that the density fluctuations show hyperuniformity [12]. We discuss that hyperuniformity of the density fluctuations is a consequence of temporal hyperuniformity of the noise, *i.e.*, the fluctuations of the noise vanish in the long-time scale $\lim_{\omega \to 0} D_q(\omega) = 0$.

Secondly, in Sec. 4, we consider the systems driven by conserving noise to investigate the effects of the spatial correlation of the noise. In previous work, Hexner and Levine have shown that for a system driven by conserving noise, the density fluctuations are highly suppressed and exhibit hyperuniformity [25], due to hyperuniformity of the noise itself [28]. Recently, Galliano *et al.* [26] have shown that the suppression of the fluctuations also stabilizes the long-range crystalline order even in two dimension, which is prohibited by the Mermin-Wagner theorem in equilibrium. Does the crystalline order also emerge in one dimension? We show that the harmonic chain driven by the conserving noise indeed possesses the crystalline order [26]. We also show that the static structure factor for small $q$ behaves as $S(q) \sim q^2$, meaning that the density fluctuations show hyperuniformity [12], as observed in previous works [25, 26]. We discuss that hyperuniformity of the density fluctuations is a consequence of spatial hyperuniformity of the noise itself, *i.e.*, $\lim_{q \to 0} D_q(\omega) = 0$.

Thirdly, in Sec. 5, we investigate a periodically driven system. For that purpose, we consider chiral active particles confined in a narrow one-dimensional channel and connected with harmonic springs [29, 30]. We show that MSD oscillates with the same frequency as that of the driving force, and the crystalline order parameter takes a finite value. We also show $S(q) \sim q^2$ for $q \ll 1$, meaning that the model shows hyperuniformity, as previously observed in chiral active matter in two dimension [20–23]. We discuss that hyperuniformity of the density fluctuations is a consequence of temporal hyperuniformity of the noise itself, *i.e.*, $\lim_{\omega \to 0} D_q(\omega) = 0$.

Finally, in Sec. 6, we consider periodically deforming particles in one dimension, which was originally introduced as a model to describe dense biological tissues [31]. The driving force of the model oscillates with the constant frequency and simultaneously has the conserving nature. The Fourier spectrum of the driving force satisfies $\lim_{q \to 0} D_q(\omega) = 0$ and $\lim_{\omega \to 0} D_q(\omega) = 0$, meaning that the driving force is spatio-temporally hyperuniform. Under the harmonic approximation, the model can be reduced to the one-dimensional harmonic chain with the oscillating natural lengths. We show that MSD oscillates with the same frequency as that of the driving force, and the crystalline order parameter takes a finite value. We also show that the model exhibits stronger hyperuniformity than those of the conserving noise and periodic driving forces: $S(q) \sim q^4$ for $q \ll 1$ [12].

The above four case studies demonstrate that the temporal and/or spatial hyperuniformity of the driving force yield hyperuniformity of the density fluctuations and can stabilize the crystalline order even in one dimension [28, 32], while the positive correlation enhances the diffusion. In Sec. 7, we summarize those results and give more quantitative discussion for the connection between the strength of hyperuniformity and existence of the crystalline order in low-dimensional systems.

## 2 Model and physical quantities

Here, we introduce the model and define a few important physical quantities.

### 2.1 Model

We consider the harmonic chain driven by the following dynamics [7]:

$$\dot{x}_j = K(x_{j+1} + x_{j-1} - 2x_j) + \xi_j(t), \quad j = 1, \cdots, N, \tag{1}$$

where $\{x_j\}_{j=1,\cdots,N}$, $\{\xi_j\}_{j=1,\cdots,N}$, $K$, and $N$ denote the positions of the particles, driving forces, spring constant, and number of particles, respectively. We impose the periodic boundary condition $x_{N+1} = x_1$. We investigate the model in the center-of-mass frame, which is, in practice, equivalent to replacing the noise in (1) as $\xi_j \to \xi_j - \sum_{k=1}^{N} \xi_k / N$. Let $R_j$ be the equilibrium position of the $j$-th particle. The dynamical equation for the displacement $u_j = x_j - R_j$ is then written as

$$\dot{u}_j = K(u_{j+1} + u_{j-1} - 2u_j) + \xi_j(t). \tag{2}$$

It is convenient to introduce the Fourier and inverse Fourier transformations of the displacement $u_j(t)$ [7, 8]:

$$u_j(t) = \frac{1}{\sqrt{N}} \sum_q \tilde{u}_q(t) e^{iqR_j}, \qquad \tilde{u}_q(t) = \frac{1}{\sqrt{N}} \sum_{j=1}^{N} u_j(t) e^{-iqR_j}, \tag{3}$$

where $q \in \left\{ \frac{2\pi k}{Na} \right\}_{k=-N/2,\cdots,N/2-1}$ if $N$ is even, and $q \in \left\{ \frac{2\pi k}{Na} \right\}_{k=-(N-1)/2,\cdots,(N-1)/2}$ if $N$ is odd. Eq. (2) is diagonalized in the Fourier space:

$$\dot{\tilde{u}}_q(t) = -\lambda_q \tilde{u}_q(t) + \tilde{\xi}_q(t), \tag{4}$$

where

$$\lambda_q = 2K\left[1 - \cos(aq)\right], \tag{5}$$

and

$$\tilde{\xi}_q(t) = \frac{1}{\sqrt{N}} \sum_{j=1}^{N} \xi_j(t) e^{-iqR_j}. \tag{6}$$

Note that $\tilde{\xi}_{q=0}(t) = 0$ in the center-of-mass frame. We assume that the mean and variance of $\tilde{\xi}_k(t)$ are given by

$$\left\langle \tilde{\xi}_q(t) \right\rangle = 0, \qquad \left\langle \tilde{\xi}_q(t)\tilde{\xi}_{q'}(t') \right\rangle = \delta_{q,-q'} D_q(t-t'). \tag{7}$$

We initialize the system with $x_j = ja$ at time $t = t_0$. Since the center of mass is fixed, the equilibrium position is given by $R_j = ja$, if the crystalline state is stable. Then, Eq. (4) can be solved directly to give

$$\tilde{u}_q(t) = \tilde{u}_q(t_0)e^{-\lambda_q(t-t_0)} + \int_{t_0}^t e^{-\lambda_q(t-s)}\tilde{\xi}_q(s)ds. \tag{8}$$

We take the limit $t_0 \to -\infty$ so that the system reaches the steady state at $t = 0$. The two point correlation in the Fourier space is then calculated as

$$\left\langle \tilde{u}_q(\omega)\tilde{u}_{q'}(\omega') \right\rangle = 2\pi \delta_{q,-q'}\delta(\omega+\omega')\frac{D_q(\omega)}{\omega^2+\lambda_q^2}. \tag{9}$$

The inverse Fourier transform w.r.t. $\omega$ yields

$$\left\langle \tilde{u}_q(t)\tilde{u}_{-q}(0) \right\rangle = \frac{1}{2\pi}\int_{-\infty}^{\infty} d\omega e^{i\omega t}\frac{D_q(\omega)}{\omega^2+\lambda_q^2} = \frac{1}{\pi}\int_0^{\infty} d\omega \frac{D_q(\omega)\cos(\omega t)}{\omega^2+\lambda_q^2}, \tag{10}$$

where we used the time-reversal symmetry of the correlation: $D_q(\omega) = D_q(-\omega)$.

## 2.2 Mean-squared displacement

Using the Parseval's identity, the mean-squared displacement in the thermodynamic limit $N \to \infty$ is calculated as follows:

$$\begin{aligned}
\text{MSD}(t) &= \frac{1}{N}\sum_{j=1}^N \left\langle \left(u_j(t)-u_j(0)\right)^2 \right\rangle \\
&= \frac{1}{N}\sum_q \left\langle \left(\tilde{u}_q(t)-\tilde{u}_q(0)\right)\left(\tilde{u}_{-q}(t)-\tilde{u}_{-q}(0)\right) \right\rangle \\
&= \frac{2}{N}\sum_q \frac{1}{\pi}\int_0^{\infty} d\omega D_q(\omega)\frac{1-\cos(\omega t)}{\omega^2+\lambda_q^2} \\
&= \frac{1}{\pi^2}\int_{-\pi/a}^{\pi/a} a dq \int_0^{\infty} d\omega D_q(\omega)\frac{1-\cos(\omega t)}{\omega^2+\lambda_q^2},
\end{aligned} \tag{11}$$

where we have replaced the summation for $q$ with an integral for $q \in (-\pi/a, \pi/a)$.

## 2.3 Order parameter

To quantify the crystalline order, we observe the Fourier component of the density at the reciprocal wave number $q = 2\pi/a$ [10]:

$$O = \frac{1}{N}\left\langle \sum_{j=1}^N e^{i\frac{2\pi}{a}x_j} \right\rangle = \left\langle e^{i\frac{2\pi u_1}{a}} \right\rangle = \left\langle \cos\left(\frac{2\pi u_1}{a}\right) \right\rangle. \tag{12}$$

The order parameter vanishes $O = 0$ for disordered liquid-like configurations, while $O > 0$ for crystals [10]. In equilibrium, the distribution of $u_1$ becomes a Gaussian. Therefore, the order parameter is calculated as $O = \exp\left[-\frac{2\pi^2 \langle u_1^2 \rangle}{a^2}\right]$. However at finite temperature, the fluctuation diverges $\langle u_1^2 \rangle \to \infty$ in the thermodynamic limit $N \to \infty$, leading to $O \to 0$ [33]. Therefore, the thermal fluctuations destroy the crystalline order in one dimension, which is consistent with the Mermin-Wagner theorem [10, 11]. Of course, the theorem does not hold in systems far from equilibrium. Indeed, in the later sections, we will show several examples of the out-of equilibrium driving forces that preserve the crystalline order even in one dimension.

## 2.4 Hyperuniformity

For perfect crystals at zero temperature, the static structure factor $S(q)$ in the limit of the small wave number vanishes: $\lim_{q \to 0} S(q) = 0$. In other words, the density fluctuations are highly suppressed for small $q$. This property is referred to as hyperuniformity [12]. In equilibrium, on the contrary, $\lim_{q \to 0} S(q)$ converges to a finite value, namely, the thermal fluctuations destroy hyperuniformity [13]. Does hyperuniformity survive under the athermal fluctuations considered in this work? To answer this question, we calculate $S(q)$ for $q \ll 1$ as follows [13]:

$$
\begin{aligned}
S(q) &= \left\langle \frac{1}{N} \left| \sum_{j=1}^{N} e^{iqx_j} \right|^2 \right\rangle \\
&\approx \frac{1}{N} \left\langle \left| \sum_{j=1}^{N} e^{iqR_j} \right|^2 \right\rangle + q^2 \left\langle \frac{1}{N} \left| \sum_{j=1}^{N} u_j e^{iqR_j} \right|^2 \right\rangle \\
&= S_0(q) + q^2 \left\langle \tilde{u}_q \tilde{u}_{-q} \right\rangle \\
&\approx q^2 \left\langle \tilde{u}_q \tilde{u}_{-q} \right\rangle,
\end{aligned}
\tag{13}
$$

where $S_0(q) = \left\langle \left| \sum_{j=1}^{N} e^{iqR_j} \right|^2 \right\rangle / N$ denotes the static structure factor of the one-dimensional lattice, which has delta peaks at $q = 2\pi n/a$, $n = 0, 1, 2, \cdots$ and can be ignored for sufficiently small but finite $q$. Eq. (13) allows us to discuss hyperuniformity from the scaling of $\left\langle \tilde{u}_q \tilde{u}_{-q} \right\rangle$ for small $q$. Note that the expansion (13) is verified only in the crystal phase. In the fluid phase, even for small $q$, $u_j$ can be very large so the expansion by $qu_j$ breaks.

The interaction potential of the harmonic chain is diagonalized in the Fourier space: $V_N = \sum_q \frac{\lambda_q}{2} \tilde{u}_q \tilde{u}_{-q}$. From the law of equipartition, one obtains $\left\langle \tilde{u}_q \tilde{u}_{-q} \right\rangle = T/\lambda_q$ in equilibrium at temperature $T$. This leads to $S(q) \approx T/(a^2 K)$ for $q \ll 1$, meaning that the thermal fluctuations destroy hyperuniformity.[1] On the contrary, hyperuniformity is often observed in systems driven by athermal fluctuations, where the law of equipartition does not hold. The simplest and well-known example is the quantum harmonic chain: $H = \sum_q \frac{\tilde{p}_q \tilde{p}_{-q}}{2} + \sum_q \frac{\lambda_q}{2} \tilde{u}_q \tilde{u}_{-q}$, where the momentum $\tilde{p}_q$ satisfies the canonical commutation relation $[\tilde{u}_q, \tilde{p}_{q'}] = i\delta_{q,-q'}\hbar$ [16, 17]. On the ground state, the distribution of $\tilde{u}_q$ is a Gaussian of zero mean and variance $\left\langle \tilde{u}_q \tilde{u}_{-q} \right\rangle = \hbar/(2\sqrt{\lambda_q})$ [34]. Therefore, the static structure factor is approximated as $S(q) \approx q\hbar/(2a\sqrt{K})$ for $q \ll 1$ [16, 17]: the model exhibits hyperuniformity $\lim_{q \to 0} S(q) = 0$. In the subsequent sections, we will show that hyperuniformity can also emerge in classical systems far from equilibrium.

---

[1]The classical harmonic chain in equilibrium does not have the crystalline order in one dimension at finite temperature [10]. However a careful treatment can justify Eq. (13) even in the fluid phase, see Ref. [13]. The same is true for quantum harmonic chain at zero temperature [17].

# 3 Temporally correlated noise

For the first example of the athermal fluctuations, we here consider the temporally correlated noise. The model allows us to understand how the time correlations of the noise yield hyperuniformity of the density fluctuations, and how these properties affect the diffusion and long-range order.

## 3.1 Settings

Here we consider the Gaussian color noise of zero mean and variance

$$\left\langle \xi_i(t)\xi_j(t')\right\rangle = \delta_{ij}D(t). \tag{14}$$

In previous work, single-file diffusion of active particles has been investigated [35]. In that case, the correlation of the noise $D(t)$ decays exponentially, which leads to the same scaling as that in equilibrium $\mathrm{MSD}(t) \sim t^{1/2}$ for $t \gg 1$ [35]. As we will see below, the scaling is altered if $D(t)$ has the power-law tail. We assume that the noise spectrum is written as

$$D(\omega) = 2T \left|\omega\right|^{-2\theta}\cos(\theta\pi), \qquad -1/2 < \theta < 1/2. \tag{15}$$

Here, the pre-factor $\cos(\theta\pi)$ has been chosen to simplify the final result, and $\theta$ is restricted to $-1/2 < \theta < 1/2$ to converge the correlation function, as we will see later. When $\theta = 0$, the model satisfies the detailed balance [7], and thus $\xi_i$ can be identified with the thermal noise at temperature $T$. For $\theta > 0$, the noise has the positive correlation that decays algebraically for large $t$: $D(t) \sim 1/t^{1-2\theta}$. On the contrary, for $\theta < 0$, $\lim_{\omega \to 0} D(\omega) = 0$, meaning that the noise fluctuations are highly suppressed in the long-time scale. Namely, the noise is temporally hyperuniform.

The power-law spectrum Eq. (15) of the noise appears for non-equilibrium systems showing self-organized criticality [36,37] and is often referred to as the $1/f$ noise [38]. The power-law spectrum also appears for the Fourier spectrum of quasi-periodic patterns. In Ref. [15], the authors showed that the Fourier spectrum of one-dimensional quasi-periodic patterns exhibits the power-law behavior for small $\omega$ with $\theta \in [-3/2, 1]$. Also, in Ref. [13], the authors argued that small perturbations to one-dimensional periodic patterns yield the power-law spectrum for $\theta \in [-1, 0]$. Therefore, the model driven by the noise with the correlation Eq. (15) would give useful insights for quasi-periodically and periodically driven systems.

The power-law spectrum (15) has been often used in the context of anomalous diffusion [39]. A single free particle driven by the noise, $\dot{x} = \xi(t)$, exhibits $\mathrm{MSD} \propto t^{1+2\theta}$ for large time $t$.

## 3.2 Mean-squared displacement

In the thermodynamic limit $N \to \infty$, the mean-squared displacement Eq. (11) is calculated as

$$\mathrm{MSD}(t) = \frac{2T}{\pi^2 \sec(\pi\theta)} \int_{-\pi/a}^{\pi/a} a \, dq \int_0^\infty d\omega \left|\omega\right|^{-2\theta} \frac{1-\cos(\omega t)}{\omega^2 + [2K(1-\cos(aq))]^2}. \tag{16}$$

The integral w.r.t $q$ can be performed as

$$\int_{-\pi/a}^{\pi/a} \frac{a \, dq}{\omega^2 + [2K(1-\cos(aq))]^2} = 2\pi \sqrt{\frac{\omega + \sqrt{\omega^2 + 16K^2}}{2\omega^3(\omega^2 + 16K^2)}} \sim \begin{cases} \frac{\pi}{\sqrt{2K\omega^3}}, & |\omega| \ll 1, \\ \frac{2\pi}{\omega^2}, & |\omega| \gg 1. \end{cases} \tag{17}$$

Using this result, one can deduce the scaling of MSD for $t \ll 1$ as follows:

$$\mathrm{MSD}(t) \sim At^{1+2\theta} \ (t \ll 1), \tag{18}$$

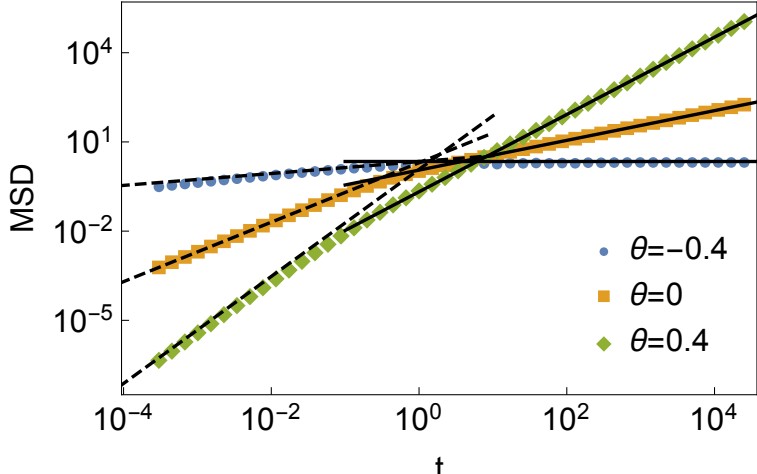

Figure 1: Mean-squared displacement of harmonic chain driven by temporally correlated noise. Markers denote exact results. Dashed and solid lines represent short and long-time asymptotic behaviors, respectively. For simplicity, we set $K = 1$ and $T = 1$.

where $A$ denotes a constant. This scaling agrees with that of a free-particle driven by the temporally correlated noise [39]. For $t \gg 1$ and $\theta > -1/4$, we get

$$\text{MSD}(t) \sim B t^{\frac{1}{2}+2\theta} \quad (t \gg 1), \tag{19}$$

where $B$ denotes a constant. For $\theta = 0$, one recovers the scaling of single-file diffusion in equilibrium MSD $\sim t^{1/2}$ [1,5]. For $\theta < -1/4$, MSD in the long time limit converges to a finite value: $\lim_{t\to\infty} \text{MSD}(t) = 2\langle u_1^2 \rangle$. We plot MSD for several $\theta$ in Fig. 1.

Eq. (19) can be understood from a simple scaling argument. To see this, we consider the continuum limit of Eq. (2):

$$\dot{u}(x,t) = K\nabla^2 u(x,t) + \xi(x,t), \tag{20}$$

where the noise correlation is given by $\langle \xi(x,t)\xi(x',t') \rangle = \delta(x-x')D(t-t')$. To analyze the model in the large spatio-temporal scale, we consider the following scaling transformations: $x \to bx$, $t \to b^{z_t}t$, $u \to b^{z_u}u$. Assuming that all terms in Eq. (20) have the same scaling dimension, we obtain $z_t = 2$ and $z_u = 1/2 + 2\theta$ [28,40]. This leads to MSD $\sim u^2 \sim b^{2z_u} \sim t^{2z_u/z_t} \sim t^{1/2+2\theta}$, which is consistent with Eq. (19).

## 3.3 Order parameter

The equal time correlation in the Fourier space is

$$\langle \tilde{u}_q \tilde{u}_{-q} \rangle = \frac{1}{\pi} \int_0^\infty \frac{D(\omega)d\omega}{\omega^2 + \lambda_q^2} = \frac{2T}{\pi \sec(\theta\pi)(\lambda_q)^{1+2\theta}} \int_0^\infty dx \frac{|x|^{-2\theta}}{x^2+1} = \frac{T}{(\lambda_q)^{1+2\theta}}. \tag{21}$$

Note that the integral converges only when $-1/2 < \theta < 1/2$. When $\theta = 0$, we recover the law of equipartition:

$$\langle \tilde{u}_q \tilde{u}_{-q} \rangle = \frac{T}{\lambda_q}. \tag{22}$$

In real space, we get

$$\langle u_1^2 \rangle = \frac{1}{N}\sum_{j=1}^N \langle u_j^2 \rangle = \frac{1}{N}\sum_q \langle \tilde{u}_q \tilde{u}_{-q} \rangle = \frac{T}{N}\sum_q \frac{1}{\lambda_q^{1+2\theta}} = \frac{T}{\pi}\int_{-\pi/a}^{\pi/a} a\,dq \frac{1}{[2K(1-\cos(aq))]^{1+2\theta}}. \tag{23}$$

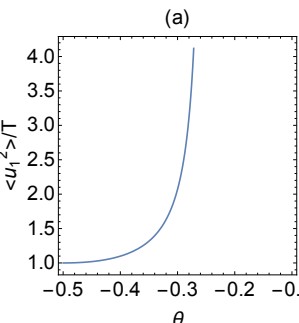
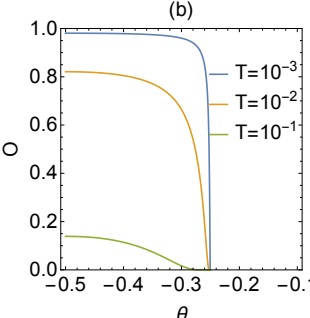

Figure 2: Physical quantities of harmonic chain driven by temporally correlated noise. (a) $\theta$ dependence of the fluctuation $\langle u_1^2 \rangle$. $\langle u_1^2 \rangle$ has a finite value for $\theta < -1/4$ and diverges at $\theta = -1/4$. (b) $\theta$ dependence of the order parameter $O$ for several temperatures. For $\theta < -1/4$, $O > 0$, while for $\theta \geq -1/4$, $O = 0$. For simplicity, we here set $K = 1$ and $a = 1$.

For $\theta < -1/4$, the integral converges to

$$\frac{\langle u_1^2 \rangle}{T} = -\frac{\theta \Gamma(-1/2 - 2\theta)}{2^{1+4\theta} K^{1+2\theta} \pi^{1/2} \Gamma(1-2\theta)}, \tag{24}$$

while for $\theta \geq -1/4$, $\langle u_1^2 \rangle \to \infty$, see Fig. 2 (a). Since $\xi_i$ is a Gaussian random number, the solution of the linear differential equation Eq. (2), $u_1$, also becomes a Gaussian random number [7]. Therefore, the order parameter can be calculated as

$$O = \left\langle e^{i\frac{2\pi u_i}{a}} \right\rangle = \exp\left[-\frac{2\pi^2}{a^2} \langle u_1^2 \rangle\right]. \tag{25}$$

We plot $O$ in Fig. 2 (b). The order parameter $O$ has a finite value for $\theta < -1/4$, meaning that the model has the long-range crystalline order even in one dimension. For $\theta > -1/4$, $O = 0$, implying that the diffusion destroys the crystalline order.

### 3.4 Hyperuniformity

In the crystal phase ($\theta < -1/4$), $S(q)$ for $q \ll 1$ is approximated as

$$S(q) \approx q^2 \left\langle \tilde{u}_q \tilde{u}_{-q} \right\rangle = T \frac{q^2}{\lambda_q^{1+2\theta}} \approx \frac{Tq^{-4\theta}}{(Ka^2)^{1+2\theta}}. \tag{26}$$

In the crystal phase, $\lim_{q\to 0} S(q) = 0$, meaning that the system is hyperuniform [12]. For $\theta < 0$, the noise spectrum satisfies $\lim_{\omega\to 0} D(\omega) = 0$, meaning that the noise is temporally hyperuniform. The above result implies that temporal hyperniformiy of the noise leads to hyperuniformiy of the density fluctuations in the crystal phase.

The above analysis is limited for $\theta > -1/2$ because the correlation Eq. (21) diverges for $\theta \leq -1/2$. This ultraviolet divergence can be removed by introducing a phenomenological cut-off to the spectrum. For that purpose, we consider the modified power-spectrum:

$$D(\omega) = \begin{cases} C\,|\omega|^{-2\theta}, & |\omega| < \omega_c, \\ 0, & \text{otherwise}, \end{cases} \tag{27}$$

where $C$ denotes a constant. The correlation Eq. (21) for $q \ll 1$ can be calculated as

$$\left\langle \tilde{u}_q \tilde{u}_{-q} \right\rangle = \frac{T}{\pi} \int_0^{\omega_c} d\omega \frac{C \, |\omega|^{-2\theta}}{\omega^2 + \lambda_q^2} \sim \frac{T}{\pi} \int_0^{\omega_c} d\omega \, |\omega|^{-2\theta-2}, \quad \theta < -1/2, \tag{28}$$

where $\omega_c$ denotes the cut-off frequency. The integral converges to a constant value for $\theta < -1/2$. Therefore, we get $S(q) \approx q^2 \left\langle \tilde{u}_q \tilde{u}_{-q} \right\rangle \sim q^2$, which is consistent with the limit $\theta \to -1/2$ of Eq. (26).

One can also investigate the effects of the power-law spatial correlation $D_q(\omega) \sim q^{-2\rho}$. Since the analysis is very parallel to that in this section, we just shortly summarize the main consequences of the power-law spatial correlation in Sec. 7. From the next sections, we shall focus on more concrete examples.

## 4 Conserving noise

In the previous section, we have observed that temporal hyperuniformity of the noise leads to hyperuniformity of the density fluctuations and also stabilizes the crystalline order even in one dimension. In this section, we shall show that spatial hyperuniformity of the noise can also yield hyperuniformity of the density fluctuations and stabilize the crystalline order [25].

### 4.1 Settings

Hyperuniformity is a phenomenon that the fluctuations of physical quantities become much smaller than what would be expected from the central limit theorem. Hyperuniformity has been reported in various systems, such as crystals, quasicrystals [13, 15, 41], and chiral active matter [21–23]. In general, the physical mechanisms causing hyperuniformity can differ depending on the details of the systems. However, Hexner and Levine have pointed out that hyperuniformity can universally appear for out-of-equilibrium systems driven by conserving noise [25]. Here, we argue that the same scenario also holds in a one-dimensional system driven by the conserving noise.

The conserving noise in the continuum limit $\xi(x, t)$ is written as $\xi(x, t) = \partial_x \eta(x, t)$, where $\eta(x, t)$ denotes another white noise. A simple implementation of this condition is

$$\xi_j(t) = \eta_j(t) - \eta_{j-1}(t), \tag{29}$$

where $\eta_j(t)$ is a Gaussian random number of zero mean and variance:

$$\left\langle \eta_i(t) \eta_j(t') \right\rangle = T \delta_{ij} \delta(t - t'). \tag{30}$$

The Fourier component of $\xi_j(t)$ satisfies

$$\left\langle \tilde{\xi}_q(t) \right\rangle = 0,$$
$$\left\langle \tilde{\xi}_q(t) \tilde{\xi}_{q'}(t) \right\rangle = 4 \delta_{q,-q'} T \left[ 1 - \cos(aq) \right] \delta(t - t') = \delta_{q,-q'} D_q(t - t'), \tag{31}$$

where

$$D_q(t) = \frac{2T \lambda_q}{K} \delta(t). \tag{32}$$

For $q \ll 1$, $\lambda_q \sim q^2$ and thus $D_q(t) \sim q^2$, meaning that the large-scale spatial fluctuations of the noise are highly suppressed. In other words, the noise is spatially hyperuniform.

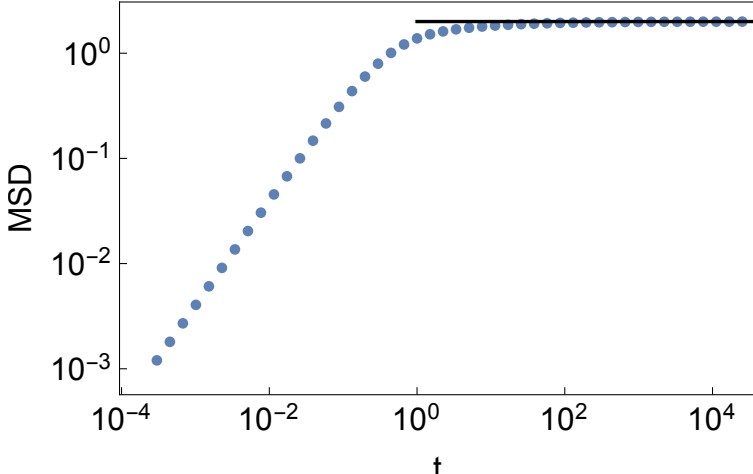

Figure 3: Mean-squared displacement of harmonic chain driven by conserving noise. Markers denote exact results. Solid lines represent long time asymptotic behavior: MSD $\sim 2\langle u_1^2 \rangle$. For simplicity, we set $K = 1$ and $T = 1$.

## 4.2 Mean-squared displacement

In the thermodynamics limit $N \to \infty$, the mean-squared displacement is calculated as

$$
\text{MSD}(t) = \frac{2T}{\pi^2 K} \int_0^\infty d\omega (1 - \cos(\omega t)) \int_{-\pi/a}^{\pi/a} a\,dq \frac{2K(1 - \cos(aq))}{\omega^2 + [2K(1 - \cos(aq))]^2} . \tag{33}
$$

We plot MSD in Fig. 3. MSD converges to a finite value in the long-time limit, $\lim_{t\to\infty} \text{MSD} = 2\langle u_1^2 \rangle$. This means that the particles are localized around their lattice positions, and thus the model is expected to have the crystalline order. Below, we confirm that this intuition is correct.

## 4.3 Order parameter

Repeating the same analysis as in Eq. (10), we get

$$
\langle \tilde{u}_q \tilde{u}_{-q} \rangle = \frac{1}{\pi} \int_0^\infty d\omega \frac{D_q(\omega)}{\omega^2 + \lambda_q^2} = \frac{2T}{K\pi} \int_0^\infty d\omega \frac{\lambda_q}{\omega^2 + \lambda_q^2} = \frac{T}{K} . \tag{34}
$$

The squared deviation from the lattice position is then calculated as

$$
\langle u_1^2 \rangle = \frac{1}{N} \sum_q \langle \tilde{u}_q \tilde{u}_{-q} \rangle = \frac{T}{K} . \tag{35}
$$

Since $\xi_j(t)$ is a Gaussian random number and the model only has the linear interactions, $u_1$ also follows the Gaussian distribution. Thus, the order parameter is

$$
O = \frac{1}{N} \left\langle \sum_{j=1}^N e^{\frac{2\pi i}{a} x_j} \right\rangle = \left\langle e^{\frac{2\pi i}{a} u_1} \right\rangle = \exp\left[ -\frac{2\pi^2 T}{a^2 K} \right] . \tag{36}
$$

The order parameter has a finite value, meaning that the model driven by the conserving noise has the crystalline order even in one dimension.

### 4.4 Hyperuniformity

Hexner and Levine argued that the density fluctuations are anomalously suppressed in systems driven by conserving noise [25]. To see this, we calculate $S(q)$ for small $q \ll 1$:

$$S(q) \approx q^2 \left\langle \tilde{u}_q \tilde{u}_{-q} \right\rangle = \frac{T}{K} q^2 \,. \tag{37}$$

$S(q)$ vanishes in the limit $q \to 0$, meaning that the large-scale density fluctuations are highly suppressed. This is the signature of hyperuniformity [12].

Overall, the above results imply that spatial hyperuniformity of the noise $\lim_{q \to 0} D_q(\omega) = 0$ yields hyperuniformity of the density fluctuations and stabilizes the long-range crystalline order even in one-dimension.

### 4.5 Mapping to Einstein model

Interestingly, the current model can be mapped into an equilibrium model. This can be seen by rewriting Eq. (4) as follows:

$$\frac{\partial \tilde{u}_q(t)}{\partial t} = -\Gamma_q \frac{\partial V_{\text{eff}}}{\partial \tilde{u}_{-q}(t)} + \tilde{\xi}_q(t) \,, \qquad \left\langle \tilde{u}_q(t) \tilde{u}_{q'}(t') \right\rangle = 2 \delta_{q,-q'} T \Gamma_q \delta(t - t') \,, \tag{38}$$

where $\Gamma_q = \lambda_q/K$ and $V_{\text{eff}} = \sum_{i=1}^{N} \frac{K}{2} u_i^2$. Eq. (38) is the equilibrium Langevin equation satisfying the detailed balance with the friction coefficient $\Gamma_q$ [7]. Then, the steady state distribution follows the Boltzmann distribution:

$$P(u_1, \cdots, u_N) = \frac{e^{-\frac{V_{\text{eff}}}{T}}}{\int \prod_{i=1}^{N} du_i e^{-\frac{V_{\text{eff}}}{T}}} \,. \tag{39}$$

This is nothing but the Einstein model consisting of $N$ independent harmonic oscillators of the same frequency $\omega = \sqrt{K}$. The Einstein model is known to exhibit hyperuniformity [13], which is consistent with Eq. (37).

## 5 Periodically driven system

In Sec. 3, we have investigated the effects of temporal hyperuniformity of the driving force $\lim_{\omega \to 0} D(\omega) = 0$. For the extreme case of temporal hyperuniformity, here we study a periodically driven system, where the Fourier spectrum of the driving force is strictly zero, $D(\omega) = 0$, for $\omega < \omega_0$.

### 5.1 Settings

Here, we consider the periodic driving force. For a concrete example, we consider chiral active particles in one dimension. Chiral active particles are particles that exhibit circular motions [30]. A popular mathematical model to describe this motion is [29]

$$
\begin{aligned}
\dot{x} &= \sqrt{2T} \cos\phi + \xi_x \,, \\
\dot{y} &= \sqrt{2T} \sin\phi + \xi_y \,, \\
\dot{\phi} &= \omega_0 + \xi_\phi \,,
\end{aligned}
\tag{40}
$$

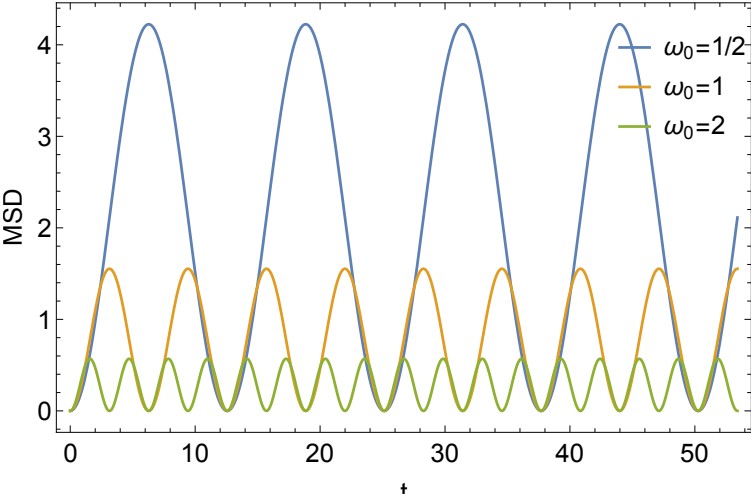

Figure 4: Mean-squared displacement of the periodically driven harmonic chain.

where $\xi_{x,y,\phi}$ denotes the noise. We are particularly interested in the limit $\xi_{x,y,\phi} \to 0$, where a chiral active particle undergoes a purely periodic motion. If the particle is confined in a one-dimensional channel along the $x$ direction, one can only consider the motion along that direction: $\dot{x} = \sqrt{2T}\cos(\omega_0 t + \phi(0))$. How does this periodic nature of the driving force affect the collective motion? To model the collective excitation of chiral active particles in one dimension, we consider the harmonic chain Eq. (1) driven by the following periodic function [32]:

$$\xi_j(t) = \sqrt{2T}\cos(\omega_0 t + \psi_j), \tag{41}$$

where $\psi_j$ denotes a random number uniformly distributed in $[0, 2\pi]$. The mean and variance of $\xi_j(t)$ are then given by

$$\begin{aligned}\langle \xi_j(t) \rangle &= 0, \\ \langle \xi_i(t)\xi_j(t') \rangle &= \delta_{ij} T D(t - t'),\end{aligned} \tag{42}$$

where $D(t) = \cos(\omega_0 t)$. The noise spectrum $D(\omega) = \pi\delta(|\omega| - \omega_0)$ vanishes in the limit of the small frequency: $\lim_{\omega \to 0} D(\omega) = 0$. Thus the noise is temporally hyperuniform. When $\omega_0 = 0$, $\xi_j(t) = \sqrt{2T}\cos\psi_j$ plays the role of the random field and destroys the long-range order in $d \leq 4$ as predicted by Imry and Ma [42]. What will happen when $\omega_0 \neq 0$?

## 5.2 Mean-squared displacement

By using Eq. (11), MSD is calculated as

$$\text{MSD}(t) = 2[1 - \cos(\omega_0 t)]\langle u_1^2 \rangle. \tag{43}$$

MSD oscillates with the frequency of the driving force $\omega_0$, see Fig. 4. The fluctuation around the lattice position $\langle u_1^2 \rangle$ is calculated as

$$\langle u_1^2 \rangle = \frac{T}{\pi} \int_0^{2\pi/a} a dq \frac{1}{\omega_0^2 + [2K(1 - \cos(aq))]^2}, \tag{44}$$

which has a finite value for $\omega_0 \neq 0$ and diverges at $\omega_0 = 0$, see Fig. 5. Therefore, the model is expected to have the crystalline order for $\omega_0 \neq 0$.

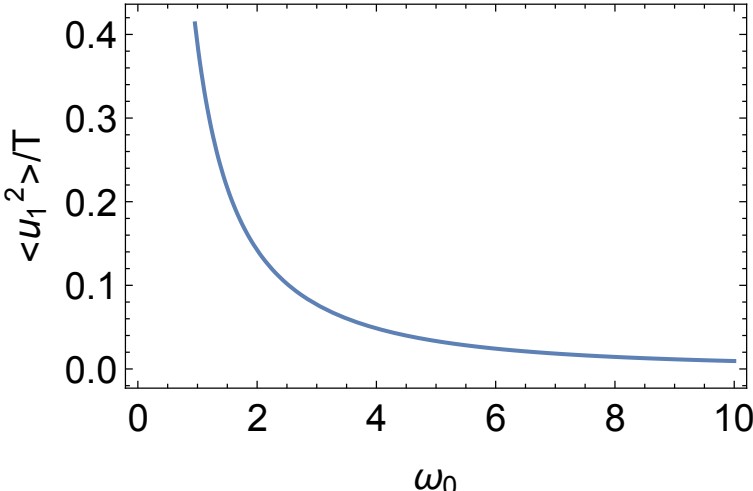

Figure 5: $\left\langle u_1^2 \right\rangle$ of the priodically driven harmonic chain. $\left\langle u_1^2 \right\rangle$ has a finite value for $\omega_0 \neq 0$ and diverges in the limit $\omega_0 \to 0$.

## 5.3 Order parameter

Because the current driving force Eq. (41) is not a Gaussian random variable, one can not easily calculate $O$. Nevertheless, one can prove the existence of the crystalline order by using the following inequality:[2]

$$O = \left\langle e^{i\frac{2\pi u_1}{a}} \right\rangle = \left\langle \cos\left(\frac{2\pi u_1}{a}\right) \right\rangle \geq 1 - \frac{2\pi^2}{a^2}\left\langle u_1^2 \right\rangle. \tag{45}$$

Since dynamics Eq. (2) does not depend on $a$, whether or not the crystalline order exists is also independent of $a$. So, we chose $a$ so that $a > \sqrt{2\pi^2 \left\langle u_1^2 \right\rangle}$. Then Eq. (45) leads to $R > 0$, meaning that the model has the crystalline order even in one dimension.

## 5.4 Hyperuniformity

Chiral active particles are known to exhibit hyperuniformity in two dimension [20–23]. Does one-dimensional system also exhibit hyperuniformity? For $q \ll 1$, the static structure factor is calculated as

$$S(q) \sim q^2 \left\langle \tilde{u}_q \tilde{u}_{-q} \right\rangle = \frac{Tq^2}{\omega_0^2 + \lambda_q^2} \sim \begin{cases} Tq^2/\omega_0^2, & \omega_0 > 0, \\ Tq^{-2}/(Ka^2)^2, & \omega_0 = 0. \end{cases} \tag{46}$$

For $\omega_0 \neq 0$, the model indeed exhibits hyperuniformity $S(q) \sim q^2$, as in chiral active matter in two dimension [20–23]. The result is also consistent with the temporally correlated noise with $\theta \leq -1/2$, see Eq. (28). This is a reasonable result because the modified power-law spectrum Eq. (27) converges to $\lim_{\theta \to -\infty} D(\omega) \propto \delta(\omega - \omega_c)$ in the limit $\theta \to -\infty$, which agrees with the Fourier-spectrum of the driving force Eq. (42). For $\omega_0 = 0$, on the contrary, one observes $S(q) \sim q^{-2}$. Therefore, $S(q)$ diverges in the limit of the small $q$. This anomalous enhancement of the large-scale density fluctuations is referred to as giant number fluctuations [43,44]. A similar power-law divergence of $S(q)$ has been previously reported for active matter in quenched random potentials [45].

---

[2]To prove the inequality $\cos(x) \geq 1 - x^2/2$, it is convenient to introduce an auxiliary function $f(x) = \cos(x) - (1 - x^2/2)$. Since $f(x)$ is an even function, it is sufficient to show $f(x) \geq 0$ for $x \geq 0$, which follows from $f(0) = 0$ and $f'(x) = -\sin(x) + x \geq 0$ for $x \geq 0$.

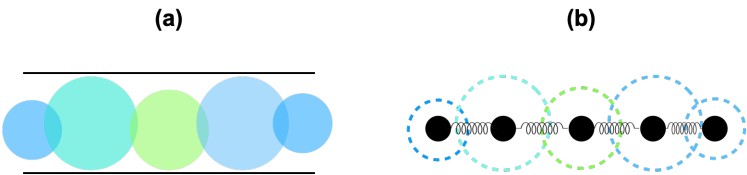

Figure 6: Schematic figures of (a) periodically deforming particles in one dimension and (b) harmonic chain where particle interactions are replaced by linear springs.

# 6 Periodically deforming particles

What will happen if the driving force is a periodic function and simultaneously conservative? To answer this question, we here consider the model introduced by Tjhung and Berthier [31].

## 6.1 Settings

Tissues are often fluidized by periodic deformations of cells [46]. To model this behavior, Tjhung and Berthier introduced periodically deforming particles [31]. The one-dimensional version of the model is written as

$$\dot{x}_j(t) = -\frac{\partial V_N}{\partial x_j}, \qquad V_N = \sum_{i<j}^{N} v(h_{ij}), \tag{47}$$

where $v(h_{ij})$ denotes the one-sided harmonic potential [47]:

$$v(h_{ij}) = \frac{Kh_{ij}^2 \Theta(-h_{ij})}{2}, \qquad h_{ij} = \left|x_i - x_j\right| - \frac{r_i(t) + r_j(t)}{2}. \tag{48}$$

Here the diameter of the $i$-th particle $r_i(t)$ oscillates with the frequency $\omega_0$ [31]:

$$r_i(t) = a + \sigma \cos(\omega_0 t + \psi_i), \tag{49}$$

where $\psi_i$ is a random number distributed uniformly in $[0, 2\pi]$. When $\omega_0 = 0$, $\sigma \cos \psi_i$ plays the role of the polydispersity, and thus, the model can not have the crystalline order.[3] The force term in Eq. (47) satisfies Newton's third law [48], which naturally leads to the conserving driving force as we will see below [25].

For sufficiently high density and small $\sigma$, the harmonic approximation would be justified, and thus the one-sided harmonic potential would be replaced by the harmonic potential (see Fig. 6):

$$v(r_{ij}) \approx \frac{Kh_{ij}^2}{2}. \tag{50}$$

Taking only the nearest neighbor interactions, one can approximate Eq. (47) as

$$\dot{x}_j \approx K(x_{j+1} + x_{j-1} - 2x_j) + K\frac{r_{j+1} - r_{j-1}}{2}. \tag{51}$$

Then, the equation of motion of the displacement $u_j$ is

$$\dot{u}_j = K(u_{j+1} + u_{j-1} - 2u_j) + \xi_j, \tag{52}$$

---

[3]For $\omega_0 = 0$, the driving force Eq. (53) becomes a quenched randomness of zero mean and variance $\langle \tilde{\xi}_q \tilde{\xi}_{q'} \rangle = T\delta_{q,-q'} \sin(aq)^2$. For $q \ll 1$, $\langle \tilde{\xi}_q \tilde{\xi}_{q'} \rangle \propto q^2 \delta_{q,-q'}$. The Imry-Ma argument [42] for the correlated disorder predicts that this type of disorder prohibits the continuous symmetry breaking for $d \leq 2$ [28]. Therefore, the polydispersity would destroy the crystalline order in one and two dimensions even without thermal fluctuations.

where

$$\xi_j(t) = \frac{K\sigma}{2} \left[ \cos(\omega_0 t + \psi_{j+1}) - \cos(\omega_0 t + \psi_{j-1}) \right]. \tag{53}$$

The mean and variance of $\tilde{\xi}_j$ are

$$\left\langle \tilde{\xi}_q(t) \right\rangle = 0,$$
$$\left\langle \tilde{\xi}_q(t)\tilde{\xi}_{q'}(t') \right\rangle = T\delta_{q,-q'}D_q(t), \tag{54}$$

where $T = (K\sigma)^2/4$ and

$$D_q(t) = \sin(aq)^2 \cos(\omega_0 t). \tag{55}$$

The noise spectrum $D_q(\omega) = \pi \sin(aq)^2 \delta(|\omega| - \omega_0)$ vanishes in the limits of small $\omega$ and/or $q$. Therefore, the noise is spatio-temporally hyperuniform.

## 6.2 Mean-squared displacement

Repeating the same analysis as in the previous sections, we get

$$\text{MSD}(t) = 2(1 - \cos(\omega_0 t))\left\langle u_1^2 \right\rangle, \tag{56}$$

where

$$\left\langle u_1^2 \right\rangle = \frac{T}{\pi} \int_0^{\pi/a} a\,dq \frac{(\sin(aq))^2}{\omega_0^2 + [2K(1 - \cos(aq))]^2}. \tag{57}$$

Eq. (56) impleis that MSD shows the periodic motion as in the case of the model considered in Sec. 5. A similar periodic motion of MSD has been previously reported by a numerical simulation of the periodically deforming particles in two dimension [49].

## 6.3 Order parameter

We plot $\left\langle u_1^2 \right\rangle$ in Fig. 7. The cage size $\left\langle u_1^2 \right\rangle$ has a finite value for $\omega_0 > 0$. In this case, using Eq. (45) and repeating the same argument as in the previous section, we can conclude that the model possesses the crystalline order. In the limit $\omega_0 \to 0$, the cage size diverges $\left\langle u_1^2 \right\rangle \to \infty$, and thus one can not prove the existence of the crystalline order. This is a natural result because when $\omega_0 = 0$, the polydispersity $\sigma \cos \psi_i$ destroys the crystalline order.

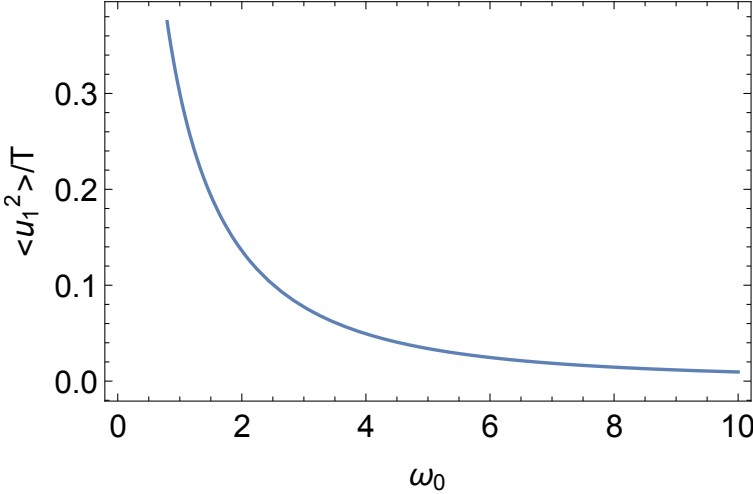

Figure 7: $\left\langle u_1^2 \right\rangle$ of periodically deforming particles. $\left\langle u_1^2 \right\rangle$ has a finite value for $\omega_0 > 0$ and diverges in the limit $\omega_0 \to 0$.

### 6.4 Hyperuniformity

For small $q \ll 1$, the static structure factor is

$$S(q) \sim q^2 \langle \tilde{u}_q \tilde{u}_{-q} \rangle = \frac{Tq^2 \sin(aq)^2}{\omega_0^2 + \lambda_q^2} \sim \begin{cases} Ta^2 q^4 / \omega_0^2, & \omega_0 > 0, \\ T/(Ka)^2, & \omega_0 = 0. \end{cases} \tag{58}$$

For $\omega_0 > 0$, we observe $S(q) \sim q^4$, which is much smaller than the result of the conserving noise Eq. (37) and periodic driving force Eq. (46). This is a consequence of the fact that the driving force Eq. (53) is a periodic function and simultaneously conservative. For $\omega_0 = 0$, $S(q)$ converges to a finite value in the limit $q \to 0$, meaning that the polydispersity destroys hyperuniformity.

Note, we used the fact that $S_0(q) = 0$ to derive Eq. (58), see Eq. (13). However, this condition is not satisfied for amorphous solids [50], and polydisperse systems studied in previous works [31, 49]. Our theory predicts that hyperuniformity is observed only in crystal phases of monodisperse systems.

## 7 Summary and discussions

### 7.1 Summary

In this work, we investigated the one-dimensional harmonic chain far from equilibrium. We considered the four types of driving forces that do not satisfy the detailed balance: (i) temporally correlated noise with power-law spectrum $D(\omega) \sim \omega^{-2\theta}$, (ii) conserving noise, (iii) periodic driving force, and (iv) periodic deformation. For the driving force (i) with $\theta > -1/4$, the model undergoes the anomalous diffusion $\mathrm{MSD}(t) \sim t^{1/2+2\theta}$. On the contrary, for the driving forces (i) with $\theta < -1/4$, and (ii)–(iv), MSD(t) remains finite. As a consequence, the crystalline order parameter has a finite value, unlike the equilibrium systems where the Mermin-Wagner theorem prohibits the long-range crystalline order in one dimension. We also discussed hyperuniformity of the density fluctuations in the crystal phase. We hope our work will stimulate further interest and progress of the long-range order [26, 28, 51–56] and hyperuniformity [13, 15, 57] in non-equilibrium low-dimensional systems.

### 7.2 Hyperuniformity

For the driving forces (i) with $\theta < 0$ and (iii), the power-spectrum of the noise vanishes in the limit of the small frequency: $\lim_{\omega \to 0} D_q(\omega) = 0$. This means that the fluctuations of the noise are highly suppressed in a long time scale, *i.e.*, the noise is temporally hyperuniform. For the driving force (ii), $\lim_{q \to 0} D_q(\omega) = 0$, implying that the noise is spatially hyperuniform. For the driving force (iv), $D_q(\omega)$ vanishes in the limits $\omega \to 0$ and/or $q \to 0$, *i.e.*, the noise is spatio-temporally hyperuniform. Our work demonstrated that these spatial and temporal hyperuniformity of the noise yield hyperuniformity of the density fluctuations [28].

For more general and quantitative discussions, we consider the spatio-temporally correlated noise whose Fourier spectrum is given by for $\omega \ll 1$ and $q \ll 1$

$$D_q(\omega) \sim \omega^{-2\theta} q^{-2\rho} . \tag{59}$$

The driving force (i) corresponds to $\rho = 0$, (ii) corresponds to $\rho = -1$ and $\theta = 0$, (iii) corresponds to $\rho = 0$ and $\theta \to -\infty$, and (iv) corresponds to $\rho = -1$ and $\theta \to -\infty$. For the noise to be hyperuniform $\lim_{q \to 0, \omega \to 0} D_q(\omega) = 0$, $\rho$ and $\theta$ should satisfy $\rho \leq 0$, $\theta \leq 0$

and $(\rho, \theta) \neq (0, 0)$. The static structure factor $S(q)$ for $q \ll 1$ is calculated as [28]

$$S(q) \approx q^2 \left\langle \tilde{u}_q \tilde{u}_{-q} \right\rangle = \frac{q^2}{\pi} \int_0^\infty d\omega \frac{D_q(\omega)}{\omega^2 + \lambda_q^2} \sim \begin{cases} q^{-2\rho - 4\theta}, & \theta > -1/2, \\ q^{2-2\rho}, & \theta \leq -1/2, \end{cases} \tag{60}$$

where the phenomenological cut-off $\omega_c$ is needed to converge the integral for $\theta \leq -1/2$, see Eq. (28). The above equation implies $\lim_{q \to 0} S(q) = 0$ if the noise is hyperuniform.

## 7.3 Anomalous diffusion of spatio-temporally correlated noise

Here we briefly discuss the anomalous diffusion of a one-dimensional system driven by the spatio-temporally correlated noise Eq. (59). For that purpose, we investigate the model in the continuum limit Eq. (20). The scaling transformations of Eq. (20), $x \to bx$, $t \to b^{z_t}$, $u \to b^{z_u} u$, lead to $z_t = 2$ and $z_u = 1/2 + 2\theta + \rho$ [28]. Then, we get the anomalous diffusion MSD $\sim t^{2z_u/z_t} \sim t^{1/2 + 2\theta + \rho}$ for $1/2 + 2\theta + \rho > 0$. For $1/2 + 2\theta + \rho < 0$, on the contrary, the diffusion is completely suppressed and the model has the long-range crystalline order.

## 7.4 Hyperuniformity and crystalline order

For the existence of the crystalline order, $\left\langle u_1^2 \right\rangle = \sum_q \left\langle \tilde{u}_q \tilde{u}_{-q} \right\rangle / N$ should remain finite in the thermodynamic limit $N \to \infty$. A necessary condition is $\left\langle \tilde{u}_q \tilde{u}_{-q} \right\rangle \propto q^{-\mu}$ with $\mu < 1$ for $q \ll 1$, which is tantamount to $S(q) \approx q^2 \left\langle \tilde{u}_q \tilde{u}_{-q} \right\rangle \sim q^{2-\mu}$ for $q \ll 1$. In other words, for the existence of the crystalline order in one dimension, the density fluctuations should exhibit sufficiently strong hyperuniformity $S(q) \sim q^\nu$ with $\nu > 1$. This condition is more stringent than in two-dimensional systems, where $\nu > 0$ is enough to stabilize the long-range crystalline order [26].

The generalization of the above argument to higher dimension $d$ is straightforward. Let $\tilde{u}(q) = \{\tilde{u}_a(q)\}_{a=1,\cdots,d}$ be the Fourier component of the displacement vector. Assuming that the system is isotropic $\left\langle \tilde{u}_a \tilde{u}_b \right\rangle = \delta_{ab} \left\langle \tilde{u}^2 \right\rangle$, one obtains $S(q) \approx |q|^2 \left\langle \tilde{u}(q) \tilde{u}(-q) \right\rangle$ for the harmonic lattice in $d$ dimension, see Ref [13]. Then, hyperuniformity $S(q) \sim |q|^\nu$ ($\nu > 0$) implies $\left\langle \tilde{u}(q) \tilde{u}(-q) \right\rangle \approx |q|^{-2} S(q) \sim |q|^{\nu-2}$. To exist the long-range crystalline order, the particles should localize around their lattice positions. In other words, the mean-squared displacement from the lattice position $\left\langle u(x)^2 \right\rangle$ should remain finite. A rough estimation of this quantity in $d$ dimension is [26]

$$\left\langle u(x)^2 \right\rangle = \frac{1}{(2\pi)^d} \int dq \left\langle \tilde{u}(q) \tilde{u}(-q) \right\rangle \sim \int_0^{q_D} dq q^{d-3+\nu}, \tag{61}$$

where $q_D$ denotes the Debye cut-off. Eq. (61) remains finite below the lower critical dimension

$$d_{\text{low}} = 2 - \nu. \tag{62}$$

Therefore, the crystalline order can exist for $\nu > 0$ in $d = 2$ [26], and $\nu > 1$ in $d = 1$. The above argument also implies that giant number fluctuations, $S(q) \sim |q|^\nu$ with $\nu < 0$, increases $d_{\text{low}}$. Using Eq. (60), we get the lower critical dimension of the crystallization of the systems driven by the spatio-temporally correlated noise

$$d_{\text{low}} = \begin{cases} 2 + 2\rho + 4\theta, & \theta > -1/2, \\ 2\rho, & \theta \leq -1/2. \end{cases} \tag{63}$$

### 7.5 Comparison with $O(n)$ model

In a previous work, we have investigated the $O(n)$ model driven by the correlated noise of the noise spectrum $D(\omega, q) \sim \omega^{-2\theta} q^{-2\rho}$ [28]. For the model-A dynamics [58], the lower critical dimension for the continuous symmetry breaking is $d_{\text{low}} = 2 + 2\rho + 4\theta$ for $\theta > -1/2$ and $d_{\text{low}} = 2\rho$ for $\theta \leq -1/2$, which agrees with Eq. (63). This is a reasonable result because the order parameter of the crystallization is a non-conservative quantity. On the contrary, since the density is a conservative quantity [58], the prediction for hyperuniformity Eq. (60) agrees with that of the model-B dynamics of the $O(n)$ model. As a consequence, the relation between hyperuniformity and $d_{\text{low}}$, Eq. (62), is not consistent with ether the model-A and model-B dynamics of the $O(n)$ model [28]. This result highlights an essential difference between the crystallization of particle systems and ferromagnetic phase transition of lattice spin systems. Further studies would be beneficial to elucidate the similarities and differences of these models.

### 7.6 Does fluid-solid transition occur in one dimension?

Several non-equilibrium systems are known to exhibit phase transition in one dimension. However, to the best of our knowledge, there are still no known examples of continuous symmetry breaking in one dimension. This work provides several promising candidates.

Our analysis for the driving forces (i) for $\theta < -1/4$ and (ii)-(iv) showed that the crystalline order can emerge even in one dimension, which is prohibited in equilibrium by the Mermin-Wagner theorem. One natural question is whether the systems driven by these driving forces exhibit liquid-solid phase transitions on increasing density. For the harmonic potential studied in this manuscript, the dynamics of the relative displacement $u_j$, Eq. (2), does not depend on the lattice spacing $a$. Therefore, the qualitative behavior of the model is also density-independent. What will happen for more realistic interaction potentials, such as the Lennard-Jones potential [59], one-sided harmonic potential, Hertzian potential [47], and so on? Extensive numerical simulations of systems driven by conserving noise [25, 26], chiral active particles [29, 30], and periodically deforming particles [31] for a wide range of density would be beneficial to elucidate this point.

## Acknowledgments

We thank Y. Nishikawa and Y. Kuroda for useful comments.

**Funding information** This project has received JSPS KAKENHI Grant Numbers 23K13031.

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
