# Peer review of "Harmonic chain far from equilibrium: single-file diffusion, long-range order, and hyperuniformity"

_SciPost Physics, doi:SciPost Phys. 17, 103 (2024)_

## Round 2 · Referee Report · Anonymous (Referee 2) · 2023-12-22

Strengths

This paper describes three situations in which a one-dimensional system of particles unable to overcome one another develop either long range order, or long-range correlations with hyperuniformity.
The article is very well written, very thorough and interesting.
It may have connections with the behavior of active particle systems in confined spaces.

Weaknesses

Phase transitions out of equilibrium in one dimensional space are quite frequent, so this `violation' of the Mermin Wagner result is not in itself the maun point of the paper.

Report

yes
  • validity: top
  • significance: high
  • originality: high
  • clarity: top
  • formatting: perfect
  • grammar: excellent

Author:  Harukuni Ikeda  on 2024-07-12  [id 4614]

(in reply to Report 2 on 2023-12-22)

Dear Referee, Thank you very much for reviewing my paper. I am very happy to see that the Referee has a high opinion of my work. I have carefully studied your reports and have revised the paper. I believe that I have fully answered your questions. In what follows, I will reply to your comments separately and explain major changes made in the revision. All the changes in the revision are highlighted in the file diff.pdf. I hope and believe that you will find the revised version acceptable for publication in SciPost. Sincerely yours,

Harukuni Ikeda

This paper describes three situations in which a one-dimensional system of particles unable to overcome one another develop either long range order, or long-range correlations with hyperuniformity.

The article is very well written, very thorough and interesting.

It may have connections with the behavior of active particle systems in confined spaces.

Thank you very much for your careful reading and positive evaluations of our work.

Phase transitions out of equilibrium in one dimensional space are quite frequent, so this `violation' of the Mermin Wagner result is not in itself the maun point of the paper.

We are sorry that our motivation has not been explained clearly in the previous version of the manuscript.

As mentioned by the referee, several non-equilibrium one-dimensional systems are known to exhibit phase transition. However, to the best of our knowledge, there are still no known examples of continuous symmetry breaking in one dimension. So, we believe that the violation of the Mermin Wagner result in one dimension is sufficiently novel.

To clarify this point, we added the following sentence at the beginning of section 7.6:

“Several non-equilibrium systems are known to exhibit phase transition in one dimension. However, to the best of our knowledge, there are still no known examples of continuous symmetry breaking in one dimension. This work provides several promising candidates.”

Attachment:

diff_hsm2YYc.pdf

---

## Round 2 · Referee Report · Anonymous (Referee 1) · 2023-12-22

Report

This manuscript describes some interesting theoretical results for a one-dimensional system of particles, interacting via harmonic potentials, and subjected to different kinds of driving or noise forces. The main questions to be addressed are whether long-ranged positional order survives the introduction of noise, and what is the nature of large-scale density fluctuations in the resulting steady state.

Several different types of driving/noise forces are considered, which allows some general trends to be identified. Overall, I think the manuscript should be suitable for sciPost after revision. However, there are some aspects of the presentation that need more precision before the manuscript can be accepted.

The most important point is that some aspects of the setup (Section 2.1) need to be more precise. The author does not give any initial conditions for the particles, to supplement equation (1). Related to this, I believe that equation (10) is valid only if the system is already in its steady state at time $t=0$. From this it seems that the angle brackets in (10) etc should represent a steady-state dynamical average. Is this correct? Full details are needed here because there are some subtleties with these kinds of analysis, see the following two points.

The author also writes that the "equilibrium position" of particle $j$ is $R_j=ja$ but since the boundaries are periodic, taking $R_j=(j-c)a$ would be an equally appropriate choice for any $c$ between $0$ and $1$. If the system supports long-ranged positional order, this means that the particles will converge to "equilibrium" positions that depend on the initialization of the system, hence $c$ also depends on the initial condition. So it is not acceptable to omit the details of initialization. (If $c$ is not zero, this also gives a phase to the order parameter $R$ in (12).)

In addition, the author takes $q=2\pi k/(Na)$ with $k=1,2,\dots,N$ so that $q$ is always positive, but then equations like (7) include $\delta_{q,-q'}$, which only makes sense if they allow $q<0$, for example by indexing $k$ symmetrically from $-N/2$ to $N/2-1$. Whichever choice is made, there is an underlying physical question about the center-of-mass mode (either $k=N$ in the author's convention or $k=0$ in the symmetric convention ). Since this mode has no restoring force ($\lambda_q=0$), it may be that the system never converges to any steady state, in which case the steady state averages mentioned above are not well-defined. A related issue is that for the simplest possible case of equilibrium dynamics with finite $N$, the scaling of the MSD as $t^{1/2}$ will cross over at large enough times to diffusive scaling, MSD $\sim t/N$, because the center of mass of the particles will undergo free diffusion. I think that several of these problems might be avoided by working in the center-of-mass frame, which should be equivalent to setting $D_q=0$ for the center-of-mass mode, from the start. There would be other options too.

If these problems are fixed then whole study will have a solid mathematical foundation. This is needed in order to properly evaluate some of the later analysis (see for example point 2 in the numbered list below).

Requested changes

1- Clarify the setup of the problem, as described in the report above.

2- What is the size of the terms that are neglected in the first approximate equality in equation (13)? Under what circumstances is it consistent to truncate this expansion? (The required assumption seems to be that $qu_j$ is small compared to unity. Even if $q$ is small, could it be that $u_j$ is very large so the expansion breaks? This step may be ok in systems with long-ranged positional order it should be justified carefully.)

3- at the end of Sec 3.1 the discussion of "hyperuniform in time" is not very clear. I would suggest to compare with a very simple system which is a single particle with position $X_j$, moving as

$$ \dot X_j = \xi_j $$
where $\xi_j$ has noise spectrum (15). I assume that one also gets anomalous scaling of the MSD in this case, depending on $\omega$. This may allow the author to connect their results with previous work on subdiffusion or fractional diffusion.

4- in eq(15), surely the sec in the denominator would be more appropriate as a cosine in the numerator.

5- please give a reference (or more detailed justification) for the inequality in (45).

6- it is unfortunate to use $R_j$ for the "equilibrium" position of particle $j$ and then later to use $R$ for the order parameter in (12).

7- In the Introduction. Please clarify the following points. First: the Mermin-Wagner theorem only applies if interactions are short-ranged. Second: the fact that particles cannot bypass one another is only true if the interactions are strong enough. Third: it is not necessarily true that fluctuating hydrodynamics is limited to low densities, see for example the macroscopic fluctuation theory of Bertini et al, which is valid at all densities in lattice models.

  • validity: ok
  • significance: good
  • originality: good
  • clarity: ok
  • formatting: good
  • grammar: excellent

Author:  Harukuni Ikeda  on 2024-07-12  [id 4613]

(in reply to Report 1 on 2023-12-22)
Category:
answer to question

Dear Referee, Thank you very much for reviewing my paper. I am very happy to see that the Referee has a high opinion of my work. I have carefully studied your reports and have revised the paper. I believe that I have fully answered your questions. In what follows, I will reply to your comments separately and explain major changes made in the revision. All the changes in the revision are highlighted in the file diff.pdf. I hope and believe that you will find the revised version acceptable for publication in SciPost. Sincerely yours,

Harukuni Ikeda

This manuscript describes some interesting theoretical results for a one-dimensional system of particles, interacting via harmonic potentials, and subjected to different kinds of driving or noise forces. The main questions to be addressed are whether long-ranged positional order survives the introduction of noise, and what is the nature of large-scale density fluctuations in the resulting steady state.

Several different types of driving/noise forces are considered, which allows some general trends to be identified. Overall, I think the manuscript should be suitable for sciPost after revision.

Thank you very much for your careful reading and positive evaluations of our work.

However, there are some aspects of the presentation that need more precision before the manuscript can be accepted.

The most important point is that some aspects of the setup (Section 2.1) need to be more precise. The author does not give any initial conditions for the particles, to supplement equation (1). Related to this, I believe that equation (10) is valid only if the system is already in its steady state at time t=0. From this it seems that the angle brackets in (10) etc should represent a steady-state dynamical average. Is this correct? Full details are needed here because there are some subtleties with these kinds of analysis, see the following two points.

Thank you for the question. As mentioned by the referee, the boundary conditions were not explicitly stated in the previous manuscript.

In this study, we set the boundary condition $\tilde{u}_q(\pm\infty)=0$ to simplify the Fourier transformation. We also assume that at finite time, $\tilde{u}_q(t)$ reaches a steady state independent from the boundary condition. To clarify this point, we have added the following sentence to the paragraph above equation (8) in the revised manuscript.

"We impose the boundary condition $\tilde{u}_q(\pm \infty)$ and assume that $\tilde{u}_q(t)$ reaches a steady state independent from the boundary condition at finite $t$.”

Also, we added the following sentence below equation (6).

“Note that $\tilde{\xi}_{q=0}(t)=0$ in the center-of-mass frame.”

The author also writes that the "equilibrium position" of particle j is $R_j=ja$ but since the boundaries are periodic, taking $R_j=(j-c)a$ would be an equally appropriate choice for any $c$ between 0 and 1. If the system supports long-ranged positional order, this means that the particles will converge to "equilibrium" positions that depend on the initialization of the system, hence $c$ also depends on the initial condition. So it is not acceptable to omit the details of initialization. (If $c$ is not zero, this also gives a phase to the order parameter $R$ in (12).)

Thank you for pointing that out. To clarify the initial condition, we modify the sentence above equation (2) as follows:

“Let $a$ be the lattice constant, $R_j=ja$ be the equilibrium position of the $j$-th particle, and $u_j=x_j-R_j$ be the displacement from the equilibrium position. The dynamical equation for $u_j$is then written as”

“Let $a$ be the lattice constant. Without loss of generality, we can assume that the equilibrium position of the $j$-th particle is given by $R_j=ja$. The dynamical equation for the displacement $u_j=x_j-R_j$ is then written as”

In addition, the author takes $q=2\pi k/(Na)$ with $k=1,2,\dots,N$ so that q is always positive, but then equations like (7) include $\delta_{q,-q'}$, which only makes sense if they allow $q<0$, for example by indexing $k$ symmetrically from $-N/2$ to $N/2-1$.

Thank you for pointing that out. Following the suggestion, we modified the sentence below equation (3) as follows:

" where $q\in \left{\frac{2\pi k}{Na} \right}_{k=1,\cdots,N}$.”

“where $q\in \left{\frac{2\pi k}{Na} \right}_{k=-N/2,\cdots, N/2-1}$ if $N$ is even, and $q\in \left{\frac{2\pi k}{Na} \right}_{k=-(N-1)/2,\cdots, (N-1)/2}$ if $N$ is odd.”

We also modified the sentence below equation (11) as follows:

“where we have replaced the summation for $q\in \left{\frac{2\pi k}{Na}\right}_{k=1,\cdots, N}$ with an integral for $q\in (0,2\pi/a]$.”

“where we have replaced the summation for $q$ with an integral for $q\in (-\pi/a,\pi/a)$.”

Accordingly, we modified the integration range in Eqs. (11), (16), (17), (23), and (33) from $\int_0^{2\pi/a}$ to $\int_{-\pi/a}^{\pi/a}$.

Whichever choice is made, there is an underlying physical question about the center-of-mass mode (either $k=N$ in the author's convention or $k=0$ in the symmetric convention ). Since this mode has no restoring force ($\lambda_q=0$), it may be that the system never converges to any steady state, in which case the steady state averages mentioned above are not well-defined. A related issue is that for the simplest possible case of equilibrium dynamics with finite $N$, the scaling of the MSD as $t^{1/2}$ will cross over at large enough times to diffusive scaling, MSD $\sim t/N$, because the center of mass of the particles will undergo free diffusion. I think that several of these problems might be avoided by working in the center-of-mass frame, which should be equivalent to setting $D_q=0$ for the center-of-mass mode, from the start. There would be other options too.

Thank you for pointing that out. Following the suggestion, we decided to use the center-of-mass frame. To clarify this point, we have added the following sentence to the paragraph below equation (1) in the revised manuscript.

“We investigate the model in the center-of-mass frame, which is, in practice, equivalent to replacing the noise in (1) as $\xi_j\to \xi_j-\sum_{k=1}^N \xi_k/N$.”

1- Clarify the setup of the problem, as described in the report above.

We clarified the setup in the revised manuscript as detailed above.

2- What is the size of the terms that are neglected in the first approximate equality in equation (13)? Under what circumstances is it consistent to truncate this expansion? (The required assumption seems to be that $qu_j$ is small compared to unity. Even if q is small, could it be that $u_j$ is very large so the expansion breaks? This step may be ok in systems with long-ranged positional order it should be justified carefully.)

Thank you for the suggestion. As mentioned by the referee, the expansion by $qu_j$ may fail in the fluid phase.

To clarify this point, we added the following sentence below equation (13):

“Note that the expansion (13) is verified only in the crystal phase. In the fluid phase, even for small $q$, $u_j$ can be very large so the expansion by $qu_j$ breaks.”

I am aware that the expansion (13) can reproduce the correct result for the classical harmonic chain in equilibrium, implying that more rigorous treatments may justify the expansion even in the fluid phase; see, for instance, Appendix. B in [J. Kim and S. Torquato, PRB 97, 054105 (2018)], which demands a bit more cumbersome calculation. Since this article mainly focuses on the connection between hyperuniformity and translational order in the crystal phase, we simply removed the discussions of $S(q)$ in the fluid phase throughout the revised manuscript. The modifications are summarized in “diff.pdf”.

3- at the end of Sec 3.1 the discussion of "hyperuniform in time" is not very clear. I would suggest to compare with a very simple system which is a single particle with position $X_j$, moving as $\dot{X}_j=\xi_j$

Thank you for the suggestion. The suggested model has been studied in a previous work in the context of anomalous diffusion. A simple scaling argument leads to ${\rm MSD}\sim t^{1+2\theta}$. To clarify this point, we added the following sentence at the end of Sec. 3.1:

“The power-low spectrum (15) has been often used in the context of anomalous diffusion[38]. A single free particle driven by the noise, $\dot{x}=\xi(t)$, exhibits ${\rm MSD}\propto t^{1+2\theta}$ for large time $t$.”

4- in eq(15), surely the sec in the denominator would be more appropriate as a cosine in the numerator.

Thank you for pointing that out. We modify the equation accordingly.

5- please give a reference (or more detailed justification) for the inequality in (45).

Thank you for the suggestion. In the revised manuscript, we added the following footnote to justify equation(45):

“To prove the inequality $\cos(x)\geq 1-x^2/2$, it is convenient to introduce an auxiliary function $f(x)=\cos(x)-(1-x^2/2)$. Since $f(x)$ is an even function, it is sufficient to show $f(x)\geq 0$ for $x\geq 0$, which follows from $f(0)=0$ and $f'(x)=-\sin(x)+x\geq 0$ for $x\geq 0$.

6- it is unfortunate to use $R_j$ for the "equilibrium" position of particle j and then later to use R for the order parameter in (12).

Thank you for the suggestion. We decided to use $O$ for the order parameter in the revised manuscript.

7- In the Introduction. Please clarify the following points. First: the Mermin-Wagner theorem only applies if interactions are short-ranged.

We modified the corresponding sentence in the introduction as follows:

“However, as proved by Mermin and Wagner, the long-range order cannot exist in one and two dimensions in equilibrium”

“However, as proved by Mermin and Wagner, the long-range order cannot exist in one and two dimensions in equilibrium if interactions are short-ranged”

Second: the fact that particles cannot bypass one another is only true if the interactions are strong enough.

We modified the first sentence in the introduction as follows:

“In one-dimensional many-particle systems, the particles cannot bypass one another.”

“In one-dimensional many-particle systems, the particles cannot bypass one another if the interactions are strong enough.”

Third: it is not necessarily true that fluctuating hydrodynamics is limited to low densities, see for example the macroscopic fluctuation theory of Bertini et al, which is valid at all densities in lattice models.

Thank you for the suggestion. We modified the corresponding sentence as follows:

“So far the most of theoretical studies of hyperuniformity far from equilibrium are based on the fluctuating hydrodynamics, which can be justified only at sufficiently low densities and can not be applied in the crystal phase[…]”

“So far, most of the theoretical studies of hyperuniformity far from equilibrium have been conducted on low densities much below the crystal phase[…]”

Attachment:

diff.pdf

---

## Round 2 · Referee Report · Anonymous (Referee 3) · 2023-12-24

Report

This work theoretically studied the vibrational dynamics of 1D harmonic chains with several active noises. In particular, the author focused on the stability of the crystalline phase and the large-scale density fluctuations. The author treated four types of active noises (temporally correlated noise, center-of-mass conserving noise, periodic driving, and periodic deformation) and derived the conditions under which the long-range order survives even in 1D and the hyperuniformity of the density emerges. In the obtained formula, the hyperuniformity exponent and the lower critical dimension are expressed using the exponents in the noise spectrum.

I feel that the paper is beneficial for future theoretical, simulation, and experimental research on dense active particles. The author treated different types of models in a unified and transparent manner, enabling readers to follow the calculations easily. Despite the simplicity of the calculations, the results and the claims are general and essential. Based on this significance, I basically recommend its publication in SciPost Physics. However, before the final recommendation, I ask the author to clarify the following point.

Requested changes

1- In the Fourier transformation Eq.(3) and (6), the zero-wave number case $q=0$ is omitted. This means that the global translation mode for the displacements and noises is omitted from the analysis. On the other hand, in Sec. 4.1, the author wrote. “To preserve the center of mass, the noise should satisfy $\sum_{j=1}^N \xi_j = 0$.“ This seems strange as the author excluded the $q=0$ mode from the beginning. I would like to ask the author to describe more about the reason why the q=0 mode can be omitted in the analysis, and to motivate the center-of-mass conserving noise in a self-consistent manner.

  • validity: good
  • significance: high
  • originality: good
  • clarity: high
  • formatting: excellent
  • grammar: excellent

Author:  Harukuni Ikeda  on 2024-07-12  [id 4615]

(in reply to Report 3 on 2023-12-24)

Dear Referee, Thank you very much for reviewing my paper. I am very happy to see that the Referee has a high opinion of my work. I have carefully studied your reports and have revised the paper. I believe that I have fully answered your questions. In what follows, I will reply to your comments separately and explain major changes made in the revision. All the changes in the revision are highlighted in the file diff.pdf. I hope and believe that you will find the revised version acceptable for publication in SciPost. Sincerely yours,

Harukuni Ikeda

This work theoretically studied the vibrational dynamics of 1D harmonic chains with several active noises. In particular, the author focused on the stability of the crystalline phase and the large-scale density fluctuations. The author treated four types of active noises (temporally correlated noise, center-of-mass conserving noise, periodic driving, and periodic deformation) and derived the conditions under which the long-range order survives even in 1D and the hyperuniformity of the density emerges. In the obtained formula, the hyperuniformity exponent and the lower critical dimension are expressed using the exponents in the noise spectrum.

I feel that the paper is beneficial for future theoretical, simulation, and experimental research on dense active particles. The author treated different types of models in a unified and transparent manner, enabling readers to follow the calculations easily. Despite the simplicity of the calculations, the results and the claims are general and essential. Based on this significance, I basically recommend its publication in SciPost Physics. However, before the final recommendation, I ask the author to clarify the following point.

Thank you very much for your careful reading and positive evaluations of our work.

1- In the Fourier transformation Eq.(3) and (6), the zero-wave number case q=0

is omitted. This means that the global translation mode for the displacements and noises is omitted from the analysis. On the other hand, in Sec. 4.1, the author wrote.

“To preserve the center of mass, the noise should satisfy $\sum_{j=1}^N\xi_j=0$.”

This seems strange as the author excluded the $q=0$ mode from the beginning. I would like to ask the author to describe more about the reason why the $q=0$ mode can be omitted in the analysis, and to motivate the center-of-mass conserving noise in a self-consistent manner.

Thank you for the suggestion. The discussion in Sec. 4.1. was indeed misleading. The crucial point of Hexner and Levine’s argument is not the conservation of the center of mass but rather that the noise in the continuum limit reduces to a conserving noise $\xi(x)=\partial_x\eta(x)$.

To clarify this point, we decided to use “conserving noise” instead of “center-of-mass conserving noise” or “center-of-mass conserving dynamics” throughout the revised manuscript.

We also modified the sentence above equation (29) as follows:

“To preserve the center of mass $X\equiv \sum_{j=1}^N x_j$, the noise should satisfy $\sum_{j=1}^N \xi_j=0$.”

“A conserving noise in the continuum limit $\xi(x)$ is written as $\xi(x)=\partial_x \eta(x)$, where $\eta(x)$ denotes another white noise.”

Attachment:

diff_3wyUWaX.pdf

---

## Round 2 · Referee Report · Anonymous (Referee 4) · 2024-1-5

Strengths

1-gives a broad assessment of the effects of non-equilibrium noise inspired by a range of physical situations
2-calculations are straightforward but set out clearly
3-interesting concluding discussion about relation to model A/B dynamics

Weaknesses

1-in most scenarios considered here, hyperuniformity - one of the key aspects being studied - is "inherited" more or less directly from properties of the assumed noise
2-treatment of center-of-mass mode and implicit steady steady assumptions not very clear

Report

The paper studies the effects of various types of non-equilibrium noise on the dynamics of the harmonic chain. The scenarios considered are inspired in part by models considered elsewhere in the literature, e.g. active particles with periodically varying sizes. Due to the linearity of the problem, most quantities of interest can be calculated in closed form, given the power spectral density $D_q(\omega)$ of the Fourier modes of the noise. The author studies in particular the structure factor $S(q)$ and its behaviour for $q\to 0$, in order to detect hyperuniformity, the mean-square displacement (MSD), and its long-time limit as a probe of crystalline order.

Of the acceptance criteria for SciPost Physics, the (only) one I can see being met is "Provide a novel and synergetic link between different research areas" as the paper does draw together a range of non-equilibrium situations considered in the literature.

Requested changes

1-The treatment of the centre-of-mass mode $q=0$ is not very clear and this mode will generically show diffusive motion, breaking many of the statements on finite MSD etc at least for long times and in systems of finite size, and preventing the system from reaching a steady state as the author implicitly assumes. I would ask the author to make these points explicit and work in the center-of-mass frame, i.e. set $\tilde{u}_{q=0}(t=0)=0$ and $D_{q=0}(\omega)=0$.
2-I found the noise in Sec. 4 not as well motivated as the others - clearly there are many ways to have center-of-mass conserving noise other than the gradient-type noise considered here; knocking out the center-of-mass noise mode (see above, equivalent to subtracting the average of the $\xi_j$ from each $\xi_i$) would be an obvious one. This should be discussed.
3-The language is generally intelligible but there are a few places where the errors are conspicuous and these should be fixed, including:
On the ground state -> In the ground state
Gaussian color_ed_ noise
hyperniformiy, hyperuniformiy -> hyperuniformity
power-low -> power-law
deriving forces -> driving forces
symmetry braking -> symmetry breaking
4-In Fig. 1 the solid line for $\theta=-0.4$ looks like it is slightly decreasing, which physically it shouldn’t - please check

  • validity: high
  • significance: good
  • originality: good
  • clarity: top
  • formatting: perfect
  • grammar: good

Author:  Harukuni Ikeda  on 2024-07-12  [id 4616]

(in reply to Report 4 on 2024-01-05)

Dear Referee, Thank you very much for reviewing my paper. I am very happy to see that the Referee has a high opinion of my work. I have carefully studied your reports and have revised the paper. I believe that I have fully answered Referee’s questions. In what follows, I will reply to your comments separately and explain major changes made in the revision. All the changes in the revision are highlighted in the file diff.pdf. I hope and believe that you will find the revised version acceptable for publication in SciPost. Sincerely yours,

Harukuni Ikeda

The paper studies the effects of various types of non-equilibrium noise on the dynamics of the harmonic chain. The scenarios considered are inspired in part by models considered elsewhere in the literature, e.g. active particles with periodically varying sizes. Due to the linearity of the problem, most quantities of interest can be calculated in closed form, given the power spectral density $D_q(\omega)$ of the Fourier modes of the noise. The author studies in particular the structure factor S(q)𝑆(𝑞) and its behaviour for q→0, in order to detect hyperuniformity, the mean-square displacement (MSD), and its long-time limit as a probe of crystalline order.

Of the acceptance criteria for SciPost Physics, the (only) one I can see being met is "Provide a novel and synergetic link between different research areas" as the paper does draw together a range of non-equilibrium situations considered in the literature.

Thank you very much for your careful reading and positive evaluations of our work.

1-The treatment of the centre-of-mass mode q=0 is not very clear and this mode will generically show diffusive motion, breaking many of the statements on finite MSD etc at least for long times and in systems of finite size, and preventing the system from reaching a steady state as the author implicitly assumes. I would ask the author to make these points explicit and work in the center-of-mass frame, i.e. set $\tilde{u}_q(t=0)=0$ and $D_{q=0}(\omega)=0$.

Thank you for pointing that out. Following the suggestion, we decided to use the center-of-mass frame. To clarify this point, we have added the following sentence to the paragraph below equation (1) in the revised manuscript.

“We investigate the model in the center-of-mass frame, which is, in practice, equivalent to replacing the noise in (1) as $\xi_j\to \xi_j-\sum_{k=1}^N \xi_k/N$.”

To simplify the Fourier transform, we impose the boundary condition $\tilde{u}_q(t=\pm \infty)=0$, instead of the initial condition $\tilde{u}_q(t=0)=0$.

2-I found the noise in Sec. 4 not as well motivated as the others - clearly there are many ways to have center-of-mass conserving noise other than the gradient-type noise considered here; knocking out the center-of-mass noise mode (see above, equivalent to subtracting the average of the $\xi_j$ from each $\xi_i$) would be an obvious one. This should be discussed.

Thank you for the suggestion. The discussion in Sec. 4.1. was indeed misleading. The crucial point of Hexner and Levine’s argument is not the conservation of the center of mass but rather that the noise in the continuum limit reduces to a conserving noise $\xi(x)=\partial_x\eta(x)$.

To clarify this point, we decided to use “conserving noise” instead of “center-of-mass conserving noise” throughout the revised manuscript.

We also modified the sentence above equation (29) as follows:

“To preserve the center of mass $X\equiv \sum_{j=1}^N x_j$, the noise should satisfy $\sum_{j=1}^N \xi_j=0$.”

“A conserving noise in the continuum limit $\xi(x)$ is written as $\xi(x)=\partial_x \eta(x)$, where $\eta(x)$ denotes another white noise.”

3-The language is generally intelligible but there are a few places where the errors are conspicuous and these should be fixed, including:

On the ground state -> In the ground state

Gaussian color_ed_ noise

hyperniformiy, hyperuniformiy -> hyperuniformity

power-low -> power-law

deriving forces -> driving forces

symmetry braking -> symmetry breaking

Thank you for the suggestions. We modified the typos.

4-In Fig. 1 the solid line for θ=−0.4 looks like it is slightly decreasing, which physically it shouldn’t - please check

The solid line for $\theta=-0.4$ denotes a constant rather than a decreasing line. The appearance of a decreasing trend may be an optical illusion.

Attachment:

diff_enz9YFf.pdf

---

## Round 3 · Referee Report · Anonymous (Referee 2) · 2024-7-15

Strengths

The paper is interesting and well written.
I am satisfied with the new version

Report

The criteria are well met

Recommendation

Publish (meets expectations and criteria for this Journal)

  • validity: top
  • significance: high
  • originality: top
  • clarity: top
  • formatting: perfect
  • grammar: excellent

Author:  Harukuni Ikeda  on 2024-08-27  [id 4722]

(in reply to Report 1 on 2024-07-15)

Dear Referee,

Thank you very much for reviewing my paper. I am pleased to see your recommendation for publication.

Sincerely yours,
Harukuni Ikeda

---

## Round 3 · Referee Report · Anonymous (Referee 1) · 2024-7-19

Report

This paper is much improved from its original version. The author has answered almost all of the criticisms of my previous report. On this basis, I would recommend acceptance, subject to the comments below. (However, I have not read the other reports in detail, so if I have not checked if their criticisms have been answered.)

I do still have a small problem with the setup. Above eq(2) it is stated
"Without loss of generality, we can assume that the equilibrium position of the jth particle is given by $R_j = ja$."
This assumption is not consistent with later statements in the manuscript because the equilibrium position of the jth particle depends on the initialisation. Similarly, it is assumed below eq(7)
"that $\tilde u_q(t)$ reaches a steady state independent from the boundary condition at finite t."
which is not the case.

The point is that if the system is initialised with $x_j=ja$ (or, equivalently, $u_j=0$) then the equilibrium positions are indeed $R_j=j.a$. But if initialised with $x_j=(j+c)a$, (equivalently, u_j=c.a) then the equilibrium positions will be different.

I also do not understand why the authors assume that $\tilde u_q(\pm \infty)= 0$. (I assume here that the $\pm\infty$ refers to the behaviour as a function of time t and not as a function of omega.) Eq(4) is a first-order Langevin equation for $\tilde u_q$ so the behaviour for large positive times is fixed by the initial conditions. Hence it is not appropriate to make this assumption on $\tilde u_q(+\infty)$, which is a random quantity with a non-trivial probability distribution

All this being said, I recommmend the following:

. Initialise the system with $x_j=ja$ [or $u_j=0$] at time $t=-\tau<0$. Since the centre of mass is fixed, this ensures that if the crystal is stable then the equilibrium position R_j of the jth particle is indeed $R_j=ja$.

. Assume that tau is large enough that the system has converged to its steady state at time t==0. (This steady state is now unique because the initial condition was prescribed. Hence this assumption can always be satified.)

. Remove any assumptions on $\tilde u_q(+\infty)$, because these cannot be satisfied in general. Side comment: I don't think that any assumption is needed about "simplifying the Fourier transform", eg note that equation (4) can be solved directly in the time domain to give (for $t>-\tau$):
$$
\tilde u_q(t) = \tilde{u}_q(-\tau) + \int_{-\tau}^t e^{\lambda(s-t)} \tilde\xi(s) ds
$$
From here one can get eq(9,10) without any additional assumptions [except for (7) and the initialisation conditions].

One other small comment:

Just after eq(26), the text reads
"The large-scale fluctuations are highly suppressed. This property is referred to as hyperuniformity [12]."
Near the start of Sec 2.4, it reads
"the density fluctuations are highly suppressed for small q. This property is referred to as hyperuniformity [12]."
It is not needed to repeat the definition of hyperuniformity. (Similar statements appear in other places too, this is not needed.)

Also, the sentence "Recently Galliano et al..." in Section 4.1 is repeated almost verbatim from the introduction (page 2). Such repetition is not needed.

Recommendation

Ask for minor revision

  • validity: -
  • significance: -
  • originality: -
  • clarity: -
  • formatting: -
  • grammar: -

Author:  Harukuni Ikeda  on 2024-08-27  [id 4724]

(in reply to Report 2 on 2024-07-19)

Dear Referee, Thank you very much for reviewing my paper. I am very happy to see that the Referee has a high opinion of my work. I have carefully studied your reports and have revised the paper. I believe that I have fully answered your questions. In what follows, I will reply to your comments separately and explain major changes made in the revision. All the changes in the revision are highlighted in the file diff.pdf. I hope and believe that you will find the revised version acceptable for publication in SciPost.

Sincerely yours, Harukuni Ikeda

This paper is much improved from its original version. The author has answered almost all of the criticisms of my previous report. On this basis, I would recommend acceptance, subject to the comments below. (However, I have not read the other reports in detail, so if I have not checked if their criticisms have been answered.)

Thank you very much for your careful reading and positive evaluations of our work.

All this being said, I recommmend the following:

. Initialise the system with $x_j=ja$ [or $u_j=0$] at time $t=-\tau<0$. Since the centre of mass is fixed, this ensures that if the crystal is stable then the equilibrium position $R_j$ of the jth particle is indeed $R_j=ja$.

. Assume that tau is large enough that the system has converged to its steady state at time $t=0$. (This steady state is now unique because the initial condition was prescribed. Hence this assumption can always be satified.)

. Remove any assumptions on $\tilde{u}_q(+\infty)$, because these cannot be satisfied in general. Side comment: I don't think that any assumption is needed about "simplifying the Fourier transform", eg note that equation (4) can be solved directly in the time domain to give (for $t>-\tau$):

$\tilde{u}q(t) = \tilde{u}_q(-\tau)+\int(s)ds$}^t e^{\lambda(s-t)}\tilde{\xi

From here one can get eq(9,10) without any additional assumptions [except for (7) and the initialisation conditions].

Thank you for pointing that out. Following the suggestions, we modified Eq. (8) and the main text accordingly. The changes in the revision are highlighted in the file diff.pdf.

Just after eq(26), the text reads

"The large-scale fluctuations are highly suppressed. This property is referred to as hyperuniformity [12]."

Near the start of Sec 2.4, it reads

"the density fluctuations are highly suppressed for small q. This property is referred to as hyperuniformity [12]."

It is not needed to repeat the definition of hyperuniformity. (Similar statements appear in other places too, this is not needed.)

Thank you for pointing that out. Following the suggestion, we shortened the corresponding sentence as follows:

“The large-scale fluctuations…”

“the system is hyperuniform.”

Also, the sentence "Recently Galliano et al..." in Section 4.1 is repeated almost verbatim from the introduction (page 2). Such repetition is not needed.

Thank you for pointing that out. We removed the corresponding sentence in the revised manuscript.

Attachment:

diff.pdf

---

## Round 3 · Referee Report · Anonymous (Referee 4) · 2024-8-11

Report

The author has made the changes I suggested and on that basis I would be happy for the paper to go forward to publication. I have not checked the changes made in response to the other referees' comments.

Recommendation

Publish (meets expectations and criteria for this Journal)

  • validity: -
  • significance: -
  • originality: -
  • clarity: -
  • formatting: -
  • grammar: -

Author:  Harukuni Ikeda  on 2024-08-27  [id 4723]

(in reply to Report 3 on 2024-08-11)

Dear Referee,

Thank you very much for reviewing my paper. I am pleased to see your recommendation for publication.

Sincerely yours,
Harukuni Ikeda

---

## Round 3 · Author Response

Dear Editors,
Thank you very much for your editorial work, especially for selecting
these four referees. I am happy that the referees have high opinions of
my work. I have carefully studied the referee reports and have revised the paper. I believe that I have fully answered the referees's questions and that they will find the revised
version acceptable for publication.
Sincerely yours,
Harukuni Ikeda

---

## Round 3 · List of Changes

Here I list major changes. 1. As suggested by Referees 1 and 4, I clarified the initial conditions in the revised manuscript. 2. As suggested by Referees 1 and 4, I used the center-of-mass frame in the revised manuscript. 3. As suggested by Referee 1, I redefine the discrete Fourier transform so that the waver number to be symmetric. 4. In response to Referee1’s comments, we decided to discuss S(q) only in the solid phase. 5. As suggested by Referee 1, I added a discussion for a free particle driven by the correlated noise in Sec. 3.1. 6. As suggested by Referee 1, I used cosθ instead of 1/secθ in the revised manuscript. 7. As suggested by Referee 1, I added a footnote to explain the inequality in (45). 8. In response to Referee1’s comments, I used O to represent the order parameter instead of R. 9. I modified the introduction as suggested by Referee 1. 10. In response to Referee 2’s comments, in Sec. 7. 6 of the revised manuscript, I discussed that the continuous symmetry breaking in one dimension has not been reported so far, even far from equilibrium. 11. In response to the comment by Referees 3 and 4, we decided to use “conserving noise” to represent the noise introduced in Sec. 4, instead of “center-of-mass conserving noise”. 12. As suggested by Referee 4, we corrected the typos.

---

## Round 4 · Author Response

Dear Editors,

Thank you for your editorial work.

I have carefully studied the referee reports and have revised the paper. I believe that I have fully answered the referees's questions and that they will find the revised version acceptable for publication.

Sincerely yours,
Harukuni Ikeda

---

## Round 4 · List of Changes

1. Following Report 2's suggestion, I modified Eq. (8) and the corresponding parts of the main text.

  2. Following Report 2's suggestion, I modified a sentence just below Eq. (26) and the corresponding parts of the main text.

  3. Following Report 2's suggestion, I removed the sentence in Sec. 4. 1 starting from "Recently Galliano et al...".

---

## Editorial Decision

published